



# Analysis of cirrus cloud over the Tibetan Plateau from CALIPSO data: an altitude perspective

Feng Zhang[1,2], Qiu-Run Yu[1], Yanyu Wang[3], Qianshan He[4,5], Tiantao Cheng[6], Xiaohong Yu[7], Dongwei Liu[4], Chunhong Chen[4]

[1]Department of Atmospheric and Oceanic Sciences, Institute of Atmospheric Sciences, Fudan University, Shanghai, 200433, China

[2]Key Laboratory of Meteorological Disaster, Ministry of Education (KLME)/ Joint International Research Laboratory of Climate and Environment Change (ILCEC), Nanjing University of Information Science and Technology, Nanjing, 210044, China

[3]Shanghai Key Laboratory of Atmospheric Particle Pollution and Prevention (LAP3), Department of Environmental Science and Engineering, Institute of Atmospheric Sciences, Fudan University, Shanghai, 200433, China

[4]Shanghai Meteorological Service, Shanghai, 201199, China

[5]Shanghai Key Laboratory of Meteorology and Health, Shanghai, 201199, China

[6]Department of Atmospheric and Oceanic Sciences, Institute of Atmospheric Sciences, Fudan University, Shanghai, 200433, China

[7]Shanxi Institute of Meteorological Sciences, Taiyuan, 030000, China

*Correspondence to: Qianshan He (oxeye75@163.com)*

**Abstract.** Using the 5-year summer Cloud-Aerosol Lidar and Infrared Pathfinder Satellite Observation (CALIPSO) data, the geographical distributions of cirrus over the Tibetan Plateau have been studied according to the cloud top height. The cirrus number at the corresponding heights exhibit striking



differences over the plateau. The maximum occurrence for cirrus top below 9 km starts over the western plateau and moves up to the northern regions when cirrus is between 9-12 km. Above 12 km, the maximum occurrence of cirrus retreats to the southern fringe of the plateau. These characteristics are linked to three kinds of formation mechanisms: large-scale orographic uplift, ice particles generation

caused by gravity wave and remnants of overflow from deep convective anvils, respectively.

# 1 Introduction

Cirrus is the high-altitude ice cloud identified as one of the most uncertain components in the current understanding of the climate variability [*Rossow and Schiffer*, 1999; *Sassen and Mace*, 2002;

*Solomon et al.*, 2007]. Cirrus cloud can profoundly affect the radiative budget of the earth-atmosphere system. They scatter the incoming solar radiation (albedo effect), prevent the outgoing longwave radiation from leaving (the greenhouse effect) and reemit the infrared radiation into the space (infrared effect), depending on their optical thickness and temperature [*McFarquhar et al.,* 2000; *Zerefos et al.*, 2003; *Corti and Peter*, 2009]. Despite influencing the atmospheric heat transport, cirrus plays an essential role in the

stratosphere-troposphere exchange of trace constituents, especially water vapor [*Rosenfield et al.*, 1998]. Recently, particular interest has been paid on cirrus in the upper troposphere and lower stratosphere (UTLS), a transition region generally recognized to control the entry of troposphere air into the stratosphere [*Gettelman et al.*, 2004; *Fueglistaler et al.*, 2009; *Randel and Jensen,* 2013].

With the onset of the Asia summer monsoon (ASM), abundant anthropogenic aerosols and their

precursors are transported to the Tibetan Plateau(TP)and can be quickly conveyed to the upper



troposphere (UT), with the vertical transportation being confined by the upper-level ASM anticyclone [*Fu et al.*, 2006; *Park et al.*, 2009; *Randel et al.*, 2010]. By scrutinizing the seasonal variation of moisture and cirrus over the TP, *Gao et al.* [2003] mentioned that the mean high cloud reflectance over the TP hit its peak in April and arrived at its minimum in November. Besides, the topographic lifting over a large

barrier can boost the elevation of relatively warm and moist air, which contributes to the substantial number of cirrus clouds in March and April [*Chen and Liu*, 2005]. Apart from the aerosols and water vapor, satellite observations also suggest that cirrus clouds are connected with the outflow from deep convection which frequently occurs over the TP [*Li et al.*, 2005; *Jin*, 2006]. Therefore, the abundant aerosols and their precursors in UTLS, the topographic lifting and the deep convection activities could

act together to promote the frequent cirrus occurrence over the TP during the ASM period.

Currently, there are two leading mechanisms for the cirrus formation: deep convective detrainment and in situ formation associated with Kelvin or gravity waves as well as the synoptic-scale ascent [*Jensen et al.*, 1996; *Pfister et al.*, 2001; *Boehm and Lee*, 2003; *Immler et al.*, 2008; *Fujiwara et al.*, 2009; *He et al.*, 2012]. It is found that cirrus is directly related to the fallout and decay of the outflow

from deep convection [*Prabhakara et al.*, 1993; *Wang et al.*, 1996]. Observations show cirrus generally occur in the vicinity of convectively active areas like the tropical western Pacific or at the places with low outgoing longwave radiation (OLR) [*Winker and Trepte*, 1998; *Eguchi et al.*, 2007]. Cirrus clouds are formed when deep convection detrains hydrometeors from the planetary boundary to the upper troposphere [*Luo et al.*, 2011]. Moreover, the temperature fluctuations driven by the large-scale vertical

uplifting or atmospheric wave activities in the upper troposphere also lead to the in situ formation of cirrus [*Riihimaki and McFarlane*, 2010]. The role of the mechanisms mentioned above to the formation of cirrus



over the TP is more complex and less understood. Detailed studies discussing the possible contributions of these mechanisms based on the cirrus top height over the Plateau are rather sparse, except some ground-based Lidar observations from a fixed site, mainly Naqu (31.5°N, 92.1°E) [*He et al.*, 2012]. Knowledge of cirrus occurrence in altitude and space and their possible explanations are critical to understand the

thermal and dynamic effects of the TP and to improve the climate modeling further.

In this paper, we investigate the variation of cirrus spatial distribution over the TP from the altitude perspective. Our particular interest is to identify the dominant contributors to the formation of cirrus at different heights over the TP, and to provide the first insight into the possible mechanisms on a regional scale. In section 2, the descriptions of the data and method are presented. Section 3 provides the

geographical distribution of cirrus and discusses its relationship with the topographic height, gravity wave and deep convection. Section 4 is the summary and brief discussions.

## 2 Data and method

### 2.1 Definition of CALIPSO Cirrus Clouds and the NOAA OLR data

The Cloud-Aerosol Lidar and Infrared Pathfinder Satellite Observation (CALIPSO) mission offers

comprehensive observations of clouds and aerosols from the troposphere to the stratosphere [*Winker et al.*, 2009; *Thorsen et al.*, 2013], and it has been proved to be highly accurate and reliable in detecting cirrus clouds [*Nazaryan et al.*, 2008]. To determine the occurrence number of cirrus clouds at different heights, we use the CALIPSO cloud layer level 2 Version 4.10 data [*Vaughan et al.*, 2009], which are acquired from the LaRC Atmospheric Sciences Data Center (ASDC) at http://eosweb.larc. nasa.gov/.

With its spatial resolution of 5 km and vertical resolution of 30 m (0-8.2 km) and 60 m (8.2-20.2 km),





CALIPSO provides not only the precise identification of cirrus clouds but also a glimpse into their vertical distribution, allowing us to gain further insight into the formation mechanisms of cirrus. To focus on the characteristics of cirrus occurrence during the ASM period, we collect the five years CALIPSO data from June to August (2012-2016). The cloud layer products include the Feature Classification Flags to identify clouds and aerosols and to discriminate their species further. The CALIPSO cloud subtyping algorithm follows the cloud top pressure thresholds from the International Satellite Cloud Climatology Project (ISCCP) cloud-type classification scheme [*Rossow and Shiffer*, 1991]. In this paper, we only use the data which is verified by the CALIPSO discrimination algorithm as cirrus. (i.e., "Feature Type" parameter equals 2 and "Feature Subtype" parameter equals 6). Moreover, only data with the cloud-and-aerosol discrimination (CAD) score between 70-100 is considered in our analysis to avoid highly uncertain cloud features [*Liu et al.*, 2009].

CALIPSO original orbital daily data is calculated into grid points data with the latitude-by-longitude resolution of 1°✕2°. We select relatively fine latitude grids and coarse longitude grids because observations are available along the given CALIPSO orbit while the adjacent track is separated by ~1.6° in the longitude. The 1°✕2° box strikes a balance between a region small enough to fully depict the variation of an individual grid and large enough to collect enough numbers of observations. In this article, the definition of the TP is from 25°-45°N and 65°-105°E with the altitude higher than 3000 m [*Yan et al.*, 2016]. In the chosen spatial domain, the CALIPSO measurements are grouped into 20 lattices, and the occurrence number of each bin is the average of all orbits passing through the corresponding grid cells.

We also employ the OLR data from the National Oceanic and Atmospheric Administration (NOAA) satellites. OLR is calculated daily as the average of the daytime and nighttime measurements by




the Advanced High-Resolution Radiometer with 2.5°×2.5° resolution [*Liebmann and Smith*, 1996]. Its value has widely been acknowledged as a proxy for the convection intensity [*Das et al.*, 2011]. Typically, OLR value below 200 W m$^{-2}$ indicates deep convection [*Fujiwara et al.*, 2009] and deep convection represents regions with extensive lifting of air that may play roles in the formation of cirrus [*He et al.*, 2013].

### 2.2 Description of Reanalysis Data

Data used in the paper also includes: the Japanese 55-year Reanalysis dataset (JRA-55; 1.25°×1.25°; 37 pressure levels) [*Kobayashi et al*., 2015] and the interim European Centre for Medium-Range Weather Forecasts Re-Analysis data (ERA-Interim; 1°×1°; 37 pressure levels) [*Dee et al.*, 2011]. The study time of all the reanalysis products in this paper is June to August from 2012-2016. To ensure the data resolution will not influence our investigation, we interpolate all reanalysis datasets onto the same horizontal resolution as that of the CALIPSO bin.

The gravity wave zonal and meridional acceleration data are acquired from the Japan Meteorological Agency. *Cheng et al.* [2014] showed that JRA-55 gave the best capture of the diurnal rainfall cycle over the TP and the eastward precipitation propagation to the eastern lees among four reanalysis datasets. Besides, JRA-55 has the smallest root mean square error in the U and V wind throughout the vertical column over the plateau [*Cheng et al.*, 2014]. By comparing with ERA-Interim and NCEP, JRA-55 also displays the best correlation in thermal heating with the station data over the plateau [*Hu and Duan et al.*, 2015]. Considering that the gravity wave in the reanalysis product is obtained according to the observationally constrained horizontal wind speeds and densities [*Cohen and Boos*,



2016], JRA-55 is therefore relatively reliable and useful for studying the gravity wave activities over the TP. Moreover, *Podglajen et al*. [2014] indicated that the ECMWF data was not adequate in representing the wave disturbances, while the standard deviation of zonal wind for JRA-55 data was more obvious despite similar geographical patterns of the two different datasets [*Kawatani et al.*, 2016].

The temperature and geopotential height from ERA-Interim are also utilized in this study [*Dee et al.*, 2011]. The variables are vertically interpolated from 1000 hPa to 1 hPa as 37 pressure levels. By verifying with 3000 high-quality and independent sounding observations, the ERA-Interim data produces relatively small mean bias in temperature profiles during the TP Experiment [*Bao and Zhang et al.*, 2012]. Other studies also prove the reliability and quality of ERA-Interim temperature and geopotential height

data over the plateau [*Gerlitz et al.*, 2014].

## 3 Results

Cirrus occurrence number is the total number of profiles identified as cirrus. In order to better probe the vertical development of cirrus, cirrus occurrence events are further grouped into four types based on the cloud top height: < 9 km; 9-12 km; 12-15 km; > 15km. Fig.1 shows the distribution of cirrus

(<9km) occurrence numbers measured by CALIPSO during the 2012-2016 summer. The brown curve represents topographic height which equals 3000 m. For cirrus top altitude less than 9 km, large numbers of cirrus are observed in the central and western part of the TP with peak numbers over 200. It is noteworthy that the large value region lies within the 4500 m topographic height line (black curve), indicating an extremely close relationship between the cirrus occurrence and the altitude. Some studies

attribute the existence of cirrus with convection produced by surface heating [*Yanai et al.*, 1992; *Chen*





*and Liu*, 2005]. The TP performs as an enormous and intense heat source with strong surface diabatic

heating in summer, since the intensity of radiation cooling is not strong enough to balance the diabatic

heating [*Wu*, 1984]. With a shallow cyclonic circulation close to the TP surface and a deep anticyclonic

circulation aloft, the moist airflows can be rapidly uplifted to the upper layers and cirrus formation is

simulated. The topmost contribution to the summer TP heating originates from the latent heat, which is

almost three times as much as the sensible heat. However, the latent heat is almost negligible at high

levels over the west flank of the TP [*Duan and Wu*, 2005]. Therefore, the combination of the sensible

heat and radiation cooling results in weak subsidence above the lower troposphere, limiting the vertical

extent of cirrus over these regions. In other words, the ascending motion and the associated water vapor

evaporation due to orographical heating are responsible for the cirrus formation, but the radiation cooling

in the upper layers prohibits the vertical growth of cirrus. Therefore, the cirrus over the high topographic

height areas is concentrated below 9 km.

Fig. 2 demonstrated the spatial distribution of cirrus occurrence number with cloud top height

between 9-12 km from 2012-2016 in summer. It is obvious that the occurrence number starts to reduce

over the highland and expands towards the north and northeast of the plateau. Considering that large

values also occur at the north side out of the TP, cirrus with cloud top between 9-12 km is generated by

external forcings different from orography. The dotted regions represent places where the corresponding

gravity wave acceleration is less than 0 from JRA-55 products. Since gravity wave propagates vertically,

these negative acceleration value means that the wave upward velocity is enhancing, and the gravity wave

is becoming more active. These dotted areas are roughly consistent with cirrus occurrence maxima,

indicating that cirrus between 9-12km is closely related with the gravity wave. In the western regions of



the TP (centered around 32°N, 80°E), topography dominates the occurrence of cirrus. Therefore, there are a few numbers of cirrus above 9 km even though the gravity wave favors their growth.

Gravity wave can be triggered by a stably stratified air overflowing obstacles such as the mountain ridge or by jet stream instabilities [*Spichtinger et al.*, 2005; *Bühler*, 2014]. The eastward subtropical jet stream passes over the TP and its adjacent orography, making them one of the world's largest providers of gravity waves [*Cohen and Boos*, 2016]. On the one hand, these waves transport horizontal momentum and interact with the circulation by propagating vertically. The wave-induced vertical uplift causing the water vapor supersaturation can trigger the occurrence of cirrus [*Spichtinger et al.*, 2005]. On the other hand, the destabilization associated with gravity waves can cause the intrusion of cold air from the stratosphere into the troposphere, which is also suggested as a possible mechanism for cirrus formation [*Boehm and Verlinde*, 2000; *Immler et al.*, 2008; *Fujiwara et al.*, 2009]. The impact of waves on the homogenous ice nucleation and the combination of homogenous and heterogeneous freezing plays a dominant role in the initial stages of cirrus formation [*Spichtinger and Krämer*, 2013; *Jensen et al.*, 2016]. In addition to ice nucleation, *Jensen and Pfister* [2004] pointed that the in situ transient temperature fluctuation can boost the atmospheric dehydration efficiency and produce larger number of ice crystals along with smaller particle size, creating more cirrus events consequently. Therefore, the fluctuations both in velocities and temperature induced by gravity wave contribute to the formation of cirrus between 9-12 km. But it should be noted that the wave-induced temperature change rather than the upward motion is the primary driver for cirrus, because the wave accelerations are on the order of $\pm 1 \mathrm{ms}^{-1}$ and there is only a few differences in vertical velocities in these regimes. This feature is also in agreement with the results of *Muhlbauer et al.* [2014].



The subtropical jet stream maximum is situated at the poleward edge of the Hadley cell in the subtropical troposphere (around 200 hPa) [*Christenson et al.*, 2017]. A large negative anomaly in zonal wind speed can be found above that height of the jet maximum [*Fujinami and Yasunari*, 2001; *Kawatani et al.*, 2004]. *Cohen and Boos* [2016] mentioned that both the weakened zonal wind and the flow density above the jet stream contributed to the increase of the Froude number, which caused the deposition of momentum and the dissipation of gravity waves. The wave breaking will eliminate the occurrence of convective instability and further limit the vertical height of cirrus. As a result, the cloud top height of cirrus induced by gravity waves is located below 12 km (200 hPa), which is coincident with the height where gravity waves start to break.

The cirrus distribution with the top height between 12-15 km is portrayed in Fig. 3a. The maxima regions dramatically shift to the southern fringe of the plateau, suggesting that cirrus above 12 km over the TP are trigger by another formation mechanism. The black dots represent areas where the maximum convective outflow level is larger than 12 km, indicating that the convective outflow and detrainment occur at the height above 12 km in these regions. The altitude where the smallest potential temperature gradient locates is defined as the convective overflow level [*Pandit et al.*, 2014]. This variable explains the rapid southward shift of cirrus maxima and also makes 12 km a threshold value for different cirrus formation mechanisms over the TP, since cirrus base is mainly found near or above the convective outflow level [*Pandit et al.*, 2014]. However, it is not expected that the cirrus observed between 12-15 km will be uniformly distributed as the geographical coverage of convective outflow level, because this variable only characterizes the thermal stratification of the atmosphere and there is no similarity between the temperature and wind profiles [*Webb*, 1958]. Meanwhile, the reanalyses data still exhibits bias and



uncertainty over the TP at a regional scale. Therefore, the height of the convective outflow level calculated from ECWMF only offers a necessary condition for the uplift of cirrus, but it is not sufficient enough to ensure the occurrence of cirrus.

Deep convection is widely accepted as a key factor for cirrus formation. In order to probe the connection between the cirrus higher than 12 km and deep convection, the daily averaged OLR distribution for 2012-2016 summer is displayed in Fig. 3b. It is obvious that OLR values decrease gradually to a minimum of 200 W/m$^{-2}$ over the southern plateau, indicating strong deep convection over these regions. Satellite data also indicates the enhanced convective events appear over the southern TP during the Asia summer monsoon [*Yanai and Li*, 1994; *Luo et al.*, 2011]. Corresponding to large values

of the cirrus number with the top height between 12-15 km in Fig. 3a, the reduced OLRs are also located over the southern part of the plateau in Fig. 3b, further validating that the cirrus among 12-15 km is mainly generated by deep convection. Deep convection, which allows sufficient vertical extent to directly inject particles to the altitude near or below the tropopause, contribute to the predominance of cirrus at low OLR regions. Apart from direct anvil spreading, deep convection can indirectly involve cirrus formation due

to the radiative cooling above the deep convective clouds and the updrafts caused by pileus clouds [*Sassen et al.*, 2009]. As a result, cirrus is formed above 12 km as remnants of overflow and dissipation from deep convective anvils. It should also be mentioned that the timing of the twice-daily CALIPSO overpasses is not in sync with the period of daily OLR data. Therefore, the maxima areas of cirrus number do not agree very well with the center of low OLR as indicated by Fig.3a and b.

The cloud top upper limit for cirrus over the Plateau is 18 km as observed by lidar. However, for cloud top above 15 km, the CALIPSO lidar observations see much less cirrus over the plateau, and there

is almost no geographical variation in cirrus numbers over these regions. Therefore, their features and the corresponding mechanisms are not discussed in this paper.

To quantify the impact of the above driving forces on the cirrus formation at their corresponding heights, we further calculate their pattern correlation coefficients [*Feng et al.*, 2016]. These coefficients reveal the relationship between two variables at corresponding locations. As indicated by Table 1, topographic height determines the distribution of cirrus below 9 km with pattern correlation coefficient being 93.7%, while gravity wave acceleration influence the cirrus between 9-12 km with correlation coefficient as -68.9%. For cirrus between 12-15 km, both the convective outflow level and OLR contribute to its occurrence with pattern correlation coefficients of 77.9% and -66.6%, respectively. Besides, all of these coefficients have passed the t-test with the 99% confidence level. Therefore, the three mechanisms mentioned above are supported statistically.

## 4 Summary and discussions

In this paper, we investigated the spatial distribution of cirrus clouds over the TP in the Asia summer monsoon season with 5-year CALIPSO data (2012-2016). Remarkable differences in the distributions of cirrus occurrence numbers are found at different heights. The cirrus with cloud top altitude less than 9 km extend almost the whole western and central part of the plateau, especially over the regions with topographic height larger than 4500 m. For cirrus with the top height between 9-12 km, distinct maxima in occurrence numbers move up to the northeastern plateau and the north side of the TP. For cirrus between 12-15 km, the maxima retreat to the southern region. There are three formation mechanisms which determine the cirrus top height over the plateau and evidence is discussed as follows:

(1) The cirrus with top height below 9 km is closely tied to orography, with pattern correlation coefficient between the topographic height and the cirrus occurrence number as 93.7%. The ascending motion and the associated water vapor evaporation due to the orographical heating in summer contribute to the cirrus formation, but the radiation cooling in the upper layers prohibits further vertical growth of cirrus over the west flank of the TP.

(2) The temperature perturbation induced by gravity wave is responsible for the maxima cirrus occurrence at the corresponding locations when the cloud top is between 9-12 km. The fluctuation can boost the atmospheric dehydration efficiency and influence the ice nucleation process, generating more cirrus particles. When gravity waves start to break above the subtropical jet (~12km), the vertical extent of cirrus is limited.

(3) Large values of cirrus numbers between 12-15 km are caused by the convective blow-off mechanism. The geographical distribution pattern of cirrus is quite similar to that of the OLR with pattern correlation coefficient as -66.6%. Since OLR is a good proxy for deep convection, cirrus formation involves both the direct and indirect effects of deep convection in low OLR regions. The direct effect is particles being directly injected to heights near or below the tropopause, while the radiative cooling above the deep convective clouds and the regional updrafts via a pileus cloud contribute to the indirect effect. Moreover, the convective outflow level determines the cloud base height of cirrus from the thermal perspective.

Our research provides the first detailed analysis of how the distribution of cirrus shifts geographically over the TP from the height perspective over a regional scale. The results help to map out the thermal and dynamical structures of the atmosphere, which determine the vertical extent of cirrus at

different geographical locations over the plateau. The unique vertical distribution of cirrus over the TP indicates special features of the connection between cirrus and physical process, and they are distinct from interactions in other regions like the tropical ocean. Therefore, the phenomena discovered in this article may promote our knowledge of cirrus over the TP and provide useful information for model

simulations. Since CALIPSO crosses the equator at 0130 and 1330 local time during a day and the orbit repeat only once in 16 days, our research is limited by the sampling time and the orbiting range-resolved resolution. More precise verification of the cirrus formation mechanisms needs to combine with intensive geostationary and in-situ observations to consider the diurnal cycle.

**Author Contributions**

Feng Zhang and Qianshan He designed the study. Qiu-Run Yu, Yanyu Wang, Tiantao Cheng, Xiaohong Yu, Dongwei Liu and Chunhong Chen contributed to data analysis, numerical experiments, interpretation and paper writing. Qiu-Run Yu and Qianshan He discussed further analysis and interpreted the results. All authors contributed to improve the manuscript.

**Acknowledgments**

This study was partially supported by the National Natural Science Foundation of China (NSFC, Grant Nos. 91637101, 91537213 and 41775129), the National Key R&D Program of China (2017YFC1501405), and the Shanghai Science and Technology Committee Research Special Funds (Grant No. 16ZR1431700). The authors gratefully acknowledge NOAA/OAR/ESRL PSD, Boulder, Colorado, USA, for providing

the interpolated OLR data on their website http://www.cdc.noaa.gov/, and the Japan Meteorological





Agency for JRA-55 data on http://jra.kishou.go.jp/JRA-55/index_en.html. Thanks also go to ECMWF

and NASA for providing ERA-Interim and CALIPSO data.

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





**Table 1.** The pattern correlation coefficients between the two variables. The * represents coefficients passing the t-test at ⩾ 99% confidence level.

| pattern correlation coefficient | topographic height | gravity wave acceleration | convection outflow level | OLR |
|---|---|---|---|---|
| cirrus number (<9 km) | 93.7%* | - | - | - |
| cirrus number (9-12 km) | - | - 68.9%* | - | - |
| cirrus number(12-15 km) | - | - | 77.9%* | -66.6%* |

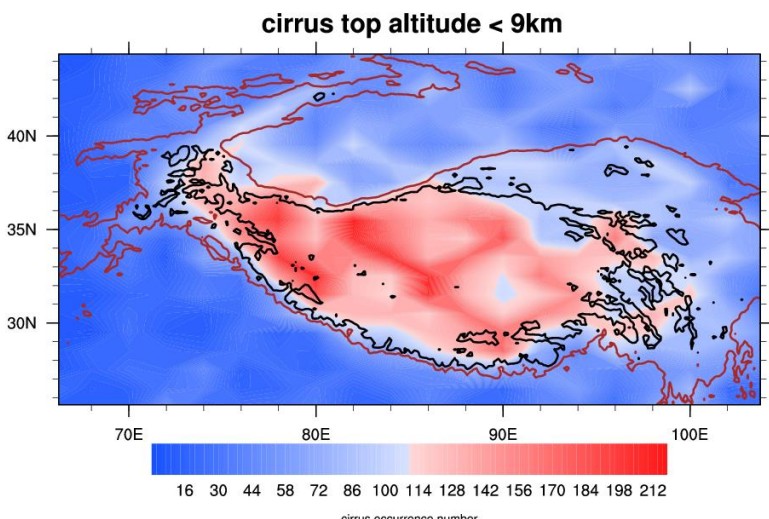

**Figure 1**. Distribution of cirrus occurrence numbers during the June-August period from 2012-2016.

The cirrus top height is below 9 km. The brown and black curves represent the topographic height of

3000 m and 4500 m, respectively.

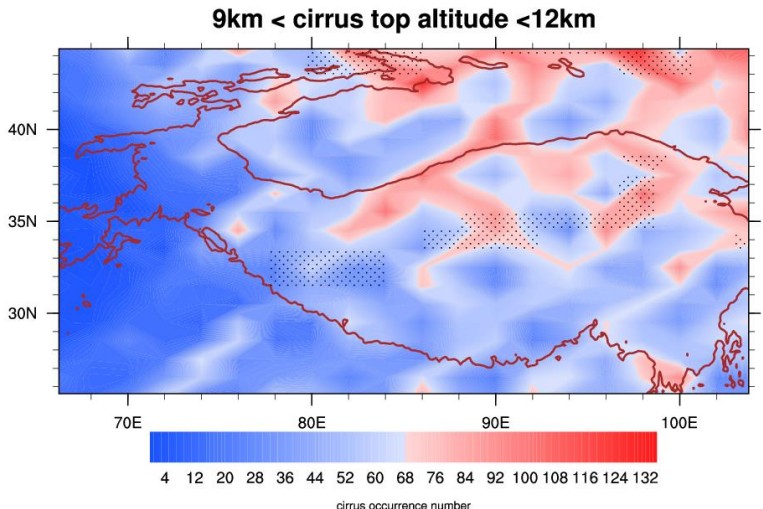

**Figure 2**. Same as Fig.1 but for cirrus top height between 9-12 km. The black dots represent regions

with negative gravity wave acceleration values.

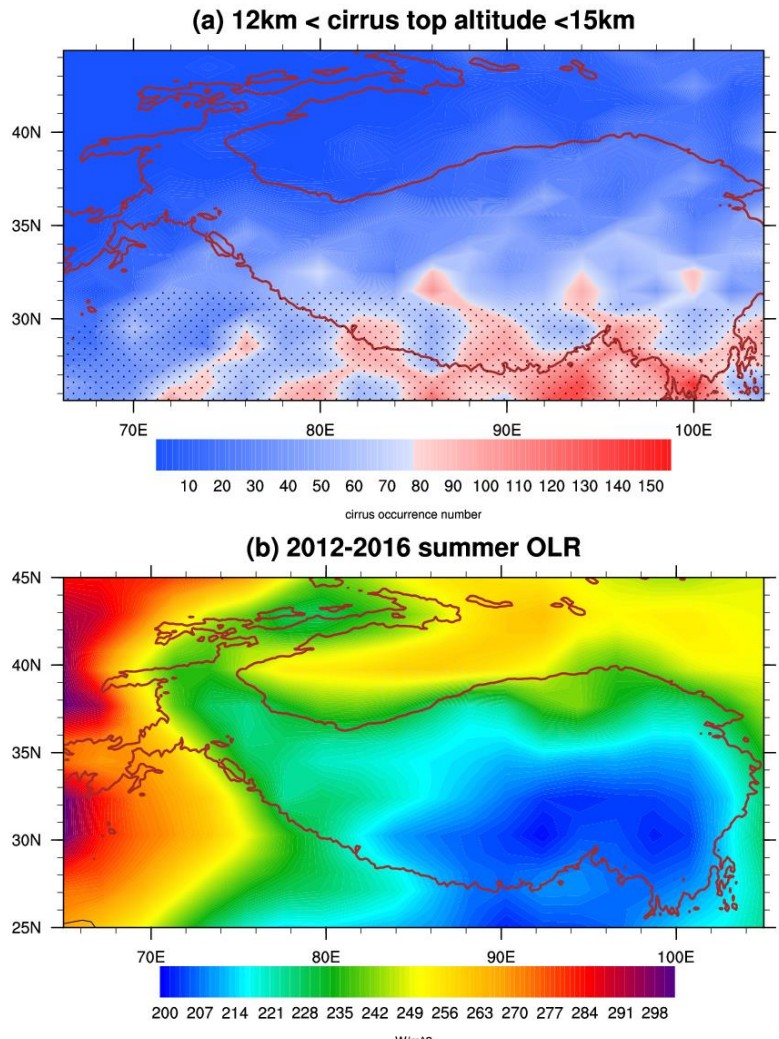

**Figure 3**. (a) Same as Fig.1 but for cirrus top height between 12-15 km. The black dots represent areas

where the convective outflow level exceeds 12 km; (b) The distribution of summer averaged OLR from

2012-2016. The brown curve represents topographic height as 3000 m.

