# Peer review of "Possible mechanisms of summer cirrus clouds over the Tibetan Plateau"

_Atmospheric Chemistry and Physics, 2019_

## Short Comment (SC1) · 8 Jan 2020

Thank the authors for providing this interesting study. I just have two short questions.

1. Indeed, the gravity waves (GWs) frequently occur around the Tibetan Plateau and the GWs possibly contribute to the cirrus formation. However, I am afraid you have not proved that the cirrus is caused, or partly caused by GWs. In Fig. 2 and Fig.3a, the "gravity wave acceleration less than 0" from JRA-55 products(if it could be considered as an index of GWs occurrence) appears at the location with and without a large number of cirrus, which indicates that the GWs may have no relevance with the occurrence of the cirrus. So I do not think it is a very convincing explanation for the cirrus formation and maybe the causality between the GWs and the cirrus should be further justified.

[Figure]

The effects of the GWs could not be verified without talking about the amplitudes. Would you please include the figures of the GWs derived from the data and method of your choice? And would you please show the amplitude of the GWs?

And from my own experience, limited by the horizontal and vertical resolution, the re-analyses, even the ERA5, could not give a nice picture of GWs in the upper troposphere and stratosphere. I think high-resolution model simulations might be necessary for this study.

P.S. in Section 3, the statements about GWs from previous studies are mixed with your results. Even though the previous studies are nicely cited, it is difficult for readers to separate your results with others'.

2. All of the key elements, e.g., the subtropical jets and the OLR in the Northern Hemisphere, the convections at the Tibetan Plateau, the occurrence of the cirrus are substantially subject to seasonal variations. I would suggest you at least separate the situations between winters and summers.

Thank the authors again for the inspiring work.

Regards, Xue

––––––––––––––––––––––––––––––

---

## Referee Comment (RC1) · Anonymous Referee #1 · 6 Feb 2020

The paper by Feng et al. investigates the geographical distributions of cirrus at three selected altitude ranges over the Tibetan Plateau using the cirrus occurrence number from CALIPSO satellite product. The authors attribute the cirrus formation at altitudes below 9 km, between 9-12 km, and above 12 km to three kinds of atmospheric dynamic factors, respectively, i.e., large-scale orographic uplift, gravity wave induced vertical uplift, and deep convection. However, while referring numerical previous studies, they did not provide stronger evidences from the data of this study to support their viewpoints. Most discussions in the current version of the paper are too descriptive to meet the high quality of an ACP research article. While admitting that the study is interesting, I do think that more in depth analyses need to be done before its acceptance.

Specific comments:

[Figure]

1. Sect. 2.1: The cirrus occurrence number from CALIPSO is used to investigate the geographical distribution of cirrus in the study. How about the geographical distribution of the effective sampling number of CALIPSO over the TP? Are there much more default values in some regions than others? Will the inhomogeneous distribution of effective sampling data result in large biases in the calculated distributions of cirrus occurrence number?

2. P7, L12 – P8, L12 and Fig. 1: I would like to see a plot showing the geographic distribution of terrain height in the region. Several variables (e.g., surface diabatic heating, radiation cooling, latent heat, sensible heat, and water vapor evaporation) are mentioned in the discussion, but none of them are displayed. Are there any signals at higher altitudes to see the influence of topographic height on cirrus? In which study and by what model is the cirrus formation simulated (stated in P8, L4-5)?

3. P8, L13 – P10, L9 and Fig. 2: It seems that the negative gravity wave acceleration cannot fully explain the distribution pattern of cirrus occurrence number shown in the figure. Could the geographical distributions of other relevant variables, such as gravity wave induced fluctuations of water vapor and temperature, be investigated? Is it possible that shallow or mid-level convection in this region play a role in the formation of cirrus?

4. P10, L10 – L12, L2 and Fig. 3: Here it might not be fully appropriate to state that deep convection is another cirrus formation mechanism (P10, L12) since atmospheric dynamics and microphysical processes in the formation of cirrus should be distinguished and described clearly. Can the difference between the timing of the CALIPSO overpasses and the period of daily OLR data fully explain the difference between the location of maximum cirrus number and the center of low OLR shown in the figure? From the geographical distribution of OLR, one can clearly see strong convection activity in most areas of eastern TP, where the cirrus occurrence number is very small. Does this indicate that the cirrus formation (occurrence number) cannot be well explained by the convection activity (OLR) at this altitude range?

5. P12, L3-11 and Table 1: What does the symbol "-" stand for in Table 1? Can the scatter plots be shown with figures?

Technical issues:

P1, L19-21: The sentence needs to be rephrased.

P1, L21: "exhibits".

P9, L16: What does "along with smaller particle size" mean? Smaller aerosol particles, or smaller cirrus particles?

P9, L19: What do you mean by saying the wave accelerations are on the order of +/- 1 m s-1? The values are too high or too low?

P10, L5: The concept of the Froude number needs to be described or explained.

P10, L12: "triggered"?

---

## Referee Comment (RC2) · Anonymous Referee #2 · 27 Mar 2020

Summary:

The authors present a manuscript that uses CALIPSO observed cloud properties, NOAA OLR data, and reanalysis data to examine the geographical distribution of the cirrus cloud over the TP. They classified the cirrus clouds into three types according to their cloud top heights with thresholds of 9 km and 12 km, then attributed the distribution patterns of cirrus corresponding to different cloud top heights to three formation mechanisms respectively, such as the orographic lifting effect, the wave enforcement, and the deep convective detrainment. I found this paper to be interesting and mostly well written and the figures to be well-constructed. I could see including it in ACP with minor improvements in the text. But I think it would be good to make the evidence stronger by adding more quantitative analysis, especially for the first and second sug-

gested mechanisms, to clarify the coupling between the contributing factors and the cirrus formation.

Specific comments:

1. For the interpretation of the first mechanism for the distribution of cirrus with cloud tops below 9 km, in Page 8 Line 7-12, the authors suggest that "the cirrus over the high topographic height areas is concentrated below 9 km" because "…the weak subsidence above the lower troposphere, limiting the vertical extent of cirrus…". However, the variables, such as the vertical motion, that used for explanation are not appeared in the provided figure. With only the topographic profiles in Fig.1, it would be difficult to imagine the inhibition effect of the radiation cooling.

2. From Page 9 to 10, the authors use three paragraphs to interpret the second mechanism, which is corresponding to the distribution of cirrus with cloud tops in the range of $9 - 12$ km. But two out of three paragraphs are totally citations of opinions from previous papers. (a) Consider reducing the citation and increasing the ratio of analysis that based on figures of the current paper. (b) Please add information to describe the wave here. Before using gravity wave acceleration to describe the intensity variation of the wave, the authors should first demonstrate the existence of gravity wave by showing amplitude or phase of the wave.

Minor comments:

1. P1L21 needs to be rephrased, consider changing to "the cirrus clouds with different cloud top heights exhibit obvious difference in their horizontal distribution over the TP…"

2. P2L1 "the maximum occurrence for cirrus top below 9 km …" is confusing in expression. "Cirrus" or "cirrus with cloud tops in range …"? please clarify this concept and rephrase the sentence.

3. P5L17 "…from 25ĚŽ-45ĚŽN and 65ĚŽ-105ĚŽE…": change to "the TP is defined

as the area that covers 25ËŽ-45ËŽN and 65ËŽ-105ËŽE" or similar expression.

4. P7L14-15 and P821: leave a space between the number and the unit "km", as i.e., "9 km".

5. Is it better to replace the colormap used for Figs 1-3a with a sequential or miscellaneous colormap? The current diverging one is not appropriate for describing monotonic trends. Besides, the font size of the "cirrus occurrence number" below the colorbar should be enlarged.

---

## Author Comment (AC1) · 23 Jun 2020

1. Indeed, the gravity waves (GWs) frequently occur around the Tibetan Plateau and the GWs possibly contribute to the cirrus formation. However, I am afraid you have not proved that the cirrus is caused, or partly caused by GWs. In Fig. 2 and Fig.3a, the "gravity wave acceleration less than 0" from JRA-55 products(if it could be considered as an index of GWs occurrence) appears at the location with and without a large number of cirrus, which indicates that the GWs may have no relevance with the occurrence of the cirrus. So I do not think it is a very convincing explanation for the cirrus formation and maybe the causality between the GWs and the cirrus should be further justified. The effects of the GWs could not be verified without talking about the amplitudes. Would you please include the figures of the GWs derived from the data and method

of your choice? And would you please show the amplitude of the GWs? And from my own experience, limited by the horizontal and vertical resolution, the re- analyses, even the ERA5, could not give a nice picture of GWs in the upper troposphere and stratosphere. I think high-resolution model simulations might be necessary for this study. P.S. in Section 3, the statements about GWs from previous studies are mixed with your results. Even though the previous studies are nicely cited, it is difficult for readers to separate your results with others'.

Response: By using Lorenz-type decomposition, the perturbation is decomposed into stationary part and transient part. The stable part is mainly caused by geographical factors, while the transient part is mainly caused by the fluctuations in the atmosphere such as gravity waves. Here is the Lorenz decomposition formula:

where overbar () and prime (′) represent the temporal mean and anomaly. Similarly, bracket (< >) and star (*) represent the spatial mean and anomaly. Thus, and are the stationary part and the transient part, respectively. When Lorenz-type decomposition is applied to the temperature field, the transient part is regarded as the temperature fluctuation which contributes to the formation of cirrus cloud. Figure 1 shows the distribution of temperature fluctuation and specific humidity, respectively. The fluctuation in temperature field is induced by gravity waves and some other convective activities. Although the temperature fluctuation in the northwest is also significant, the water vapor there is not enough to form cirrus clouds. 2. All of the key elements, e.g., the subtropical jets and the OLR in the Northern Hemisphere, the convections at the Tibetan Plateau, the occurrence of the cirrus are substantially subject to seasonal variations. I would suggest you at least separate the situations between winters and summers. Response: Thanks for your suggestion. Our study focus on the formation of cirrus in summer. It would also be interesting to explore the cirrus in winter over the TP in the future.

[Figure]

[Figure]

**Fig. 1.** distribution of temperature fluctuation and specific humidity

---

## Author Comment (AC2) · 23 Jun 2020

1. Sect. 2.1: The cirrus occurrence number from CALIPSO is used to investigate the geographical distribution of cirrus in the study. How about the geographical distribution of the effective sampling number of CALIPSO over the TP? Are there much more default values in some regions than others? Will the inhomogeneous distribution of effective sampling data result in large biases in the calculated distributions of cirrus occurrence numbers? Response: Fig. 1 shows the geographical distribution of the effectively sampled cirrus number by CALIPSO over the TP during the summertime from 2012 to 2016. The spatial resolution is $1° \times 2°$ and the sampling criteria are the same as that in the manuscript. We can tell that only two regions failed to have values. Both of them are on the outer edges of Tibet Plateau so the default values there won't affect

our study. Despite these regions, the rest of the study areas have abundant sampling numbers to allow us to gain a solid knowledge of the cirrus characteristics. The effective sampling data results are indeed inhomogeneous; however, this does not mean a large bias of the CALIPSO data. All the numbers counted in our study are quality assured. The inhomogeneity can be influenced by the CALIPSO orbit and how we set our domain resolution, but considering the large sampling numbers, this geographical inhomogeneity can reveal the reality of cirrus distribution and this is also the inspiration of our study. We want to explore what kind of mechanism triggered this geographical inhomogeneity of cirrus and the characteristic of cirrus on the Tibetan Plateau.

Figure 1. Geographical distribution of cirrus numbers effectively sampled by CALIPSO during the June-August period from 2012-2016.

Figure 2. Topographic maps over Tibetan Plateau 2. P7, L12 – P8, L12 and Fig. 1: I would like to see a plot showing the geographic distribution of terrain height in the region. Several variables (e.g., surface diabatic heating, radiation cooling, latent heat, sensible heat, and water vapor evaporation) are mentioned in the discussion, but none of them are displayed. Are there any signals at higher altitudes to see the influence of topographic height on cirrus? In which study and by what model is the cirrus formation simulated (stated in P8, L4-5)? Response: Fig. 2 shows the geographic distribution of terrain height in the region. Fig.3 shows the monthly mean surface net thermal radiation, water vapor evaporation, latent heat flux and sensible heat flux from ERA5 data, respectively. Radiative cooling is the net outgoing radiative energy flux(Sun, Sun, Zhou, Alam, & Bermel, 2017), it can be given as

Where is the thermal emission of the radiative cooler with temperature , and is the atmospheric radiation with air temperature . Here we assume the atmospheric radiation is the same in our study region, the radiative cooling is determined by the surface thermal emission , which is the upper left plot in Fig. 3. Essentially, the maximum radiative cooling region lies in the southwest of the Plateau where the terrain height exceeds 4500 m. The top right and bottom left figures show the evaporation and surface latent

heat flux, respectively. Their patterns are identical. The regions with higher altitude tend to be drier than lower altitude regions, so evaporation and surface latent heat flux is not the main contributor to the formation of cirrus below 9 km. The bottom right figure shows the surface sensible heat flux. Higher altitude region also shows strong sensible heat flux. However, the magnitude is around 70 Wm-2, which is smaller than the magnitude of surface radiative cooling (130 Wm-2). Therefore, the surface radiative cooling caused by terrain height triggered cirrus below 9 km and the surface sensible heat flux is the second contributor. Figure 3. Geographical distribution of monthly mean (a) surface radiation cooling, (b) evaporation (c) latent heat flux, and (d) sensible heat flux over Tibetan Plateau. The study period is June, July, and August from 2012 to 2016. 3. P8, L13 – P10, L9 and Fig. 2: It seems that the negative gravity wave acceleration cannot fully explain the distribution pattern of cirrus occurrence number shown in the figure. Could the geographical distributions of other relevant variables, such as gravity wave induced fluctuations of water vapor and temperature, be investigated? Is it possible that shallow or mid-level convection in this region play a role in the formation of cirrus? Response: We agree with the reviewer. The negative gravity wave acceleration cannot fully explain the distribution pattern of cirrus occurrence number. Following the classical circulation decomposition [Lorenz, 1967], the perturbation is decomposed into stationary part and transient part. The stationary part is mainly caused by geographical factors, while the transient part is mainly caused by the fluctuations in the atmosphere such as gravity waves. Here is the Lorenz decomposition formula:

where overbar () and prime (′) represent the temporal mean and anomaly. Similarly, bracket (< >) and star (*) represent the spatial mean and anomaly. Thus, and are the stationary part and the transient part, respectively. Figure 4 shows the geographical distribution of (a) transient temperature fluctuation and (b) 5-year averaged specific humidity at 250 hPa (about 11 to 12 km). There is significant temperature fluctuation at the north side of the Tibet Plateau, with a peak near 79 ° E and 41 ° N. However, the water vapor condition at 250 hPa over the western TP is too poor to form more cirrus clouds, so the cirrus clouds are concentrated in the northeast. Temperature fluctuation includes convections, gravity waves, and other atmospheric activities at different scales. Besides, the convections, the eastward subtropical upper-level jet stream passes over the TP and its adjacent orography are all likely to trigger gravity waves and intensify temperature fluctuation [Cohen and Boos, 2016]. Therefore, the fluctuations in temperature contribute to the formation of cirrus between 9-12 km.

Figure 4. Geographical distribution of (a) temperature fluctuation and (b) 5-year averaged specific humidity at 250 hPa (about 11 to 12km). 4. P10, L10 – L12, L2 and Fig. 3: Here it might not be fully appropriate to state that deep convection is another cirrus formation mechanism (P10, L12) since atmospheric dynamics and microphysical processes in the formation of cirrus should be distinguished and described clearly. Can the difference between the timing of the CALIPSO overpasses and the period of daily OLR data fully explain the difference between the location of maximum cirrus number and the center of low OLR shown in the figure? From the geographical distribution of OLR, one can see strong convection activity in most areas of eastern TP, where the cirrus occurrence number is very small. Does this indicate that the cirrus formation (occurrence number) cannot be well explained by the convection activity (OLR) at this altitude range? Response: Yes, our conclusion can be affected by the timing of the CALIPSO overpasses. CALIPSO passes our interested regions twice a day while the OLR data is daily. Moreover, OLR is reanalyzed grid data while the CALIPSO sampling number is the mean of each $1° \times 2°$ box. These two reasons can cause a mismatch between strong OLR value and small cirrus occurrence number. However, OLR is just an indicator of the deep convection. Deep convection alone cannot guarantee the formation of cirrus, and other factors such as condensation nuclei and water vapor are also needed. Therefore, the convective outflow level and OLR only offer a necessary condition for the uplift of cirrus, but it is not sufficient enough to ensure the occurrence of cirrus. As we can see from Figure 7, the convective overflow height is around 12 km in most areas of eastern TP and the OLR is below 210 Wm-2, indicating strong convection activities there. From Fig.5a and Fig. 6a, we can see water vapor is more abundant when latitude is smaller than 30N at 200 hPa. The atmospheric vertical motion and favorable water vapor condition helps the formation of cirrus above 12km.

Figure 5. Geographical distribution of (a) specific humidity anomaly and (b) temperature anomaly from monthly ERA5 data.

Figure 6. Geographical distribution of (a) specific humidity absolute anomaly and (b) temperature absolute anomaly from monthly ERA5 data.

Figure 7. Distribution of convective overflow height and OLR. 5. P12, L3-11 and Table 1: What does the symbol "-" stand for in Table 1? Can the scatter plots be shown with figures? Response: Symbol "-" stands for failing to pass the significant test. Scatter plots are less intuitive than direct correlation coefficient. Therefore, they are ignored here. Technical issues: P1, L19-21: The sentence needs to be rephrased. Response: Thank you for the suggestion. Then sentence has been rephrased as "The geographical distributions of summertime cirrus with different cloud-top heights above the Tibetan Plateau are investigated by using the 2012 - 2016 Cloud-Aerosol Lidar and Infrared Pathfinder Satellite Observation (CALIPSO) data.". P1, L21: "exhibits". Response: Corrected. P9, L16: What does "along with smaller particle size" mean? Smaller aerosol particles, or smaller cirrus particles? Response: Sorry for the misleading information. The increase of ice crystals numbers will bring the shrink of their size, so the smaller particle size means the smaller ice particle size. This sentence has been changed correspondingly. P9, L19: What do you mean by saying the wave accelerations are on the order of +/- 1 m s-1? The values are too high or too low? Response: These values are relatively low. Therefore the wave acceleration is not the only contributor. Fluctuations both in velocities and temperature-induced by gravity wave contribute to the formation of cirrus between 9-12 km. P10, L5: The concept of the Froude number needs to be described or explained. Response: Thank you. This part has been added. We have added related information into the text. P10, L12: "triggered"? Response: corrected. We appreciate Reviewer 1 very much for his constructive comments.

Please also note the supplement to this comment:
https://www.atmos-chem-phys-discuss.net/acp-2019-1000/acp-2019-1000-AC2-
supplement.pdf
* * *
[Figure]

**2012-2016 summer cirrus occurrence number distribution**

**Fig. 1.** Geographical distribution of cirrus numbers effectively sampled by CALIPSO during the June-August period from 2012-2016.

**Topographic maps**

elevation (meters)

**Fig. 2.** Topographic maps over Tibetan Plateau

**2012-2016 summer monthly mean surface variables**

[Figure]

**Fig. 3.** Geographical distribution of monthly mean (a) surface radiation cooling, (b) evaporation (c) latent heat flux, and (d) sensible heat flux over Tibetan Plateau. The study period is June, July, and Augu

[Figure]

**Fig. 4.** Geographical distribution of (a) temperature fluctuation and (b) 5-year averaged specific humidity at 250 hPa (about 11 to 12km).

[Figure]

**Fig. 5.** Geographical distribution of (a) specific humidity anomaly and (b) temperature anomaly from monthly ERA5 data.

[Figure]

**Fig. 6.** Geographical distribution of (a) specific humidity absolute anomaly and (b) temperature absolute anomaly from monthly ERA5 data.

[Figure]

**Fig. 7.** Distribution of convective overflow height and OLR.

---

## Author Comment (AC3) · 23 Jun 2020

1. For the interpretation of the first mechanism for the distribution of cirrus with cloud tops below 9 km, in Page 8 Line 7-12, the authors suggest that "the cirrus over the high topographic height areas is concentrated below 9 km" because ". . .the weak sub- sidence above the lower troposphere, limiting the vertical extent of cirrus. . .". However, the variables, such as the vertical motion, that used for explanation are not appeared in the provided figure. With only the topographic profiles in Fig.1, it would be difficult to imagine the inhibition effect of the radiation cooling. Response: Thank you for the advice. Figure 8 shows the vertical wind averaged from 80E to 90E for each latitude. The contour line is specific humidity. Here we choose to do meridional average from 80E to 90E because this is the main region of the Plateau where terrain

height exceeds 4500m. The red rectangle shows the weak subsidence below 9 km (300 hPa approximately). This part has been added in the manuscript.

Figure 8. the zonal distribution of vertical winds averaged from 80E to 90E for each latitude. The contour is specific humidity. 2. From Page 9 to 10, the authors use three paragraphs to interpret the second mecha- nism, which is corresponding to the distribution of cirrus with cloud tops in the range of 9 – 12 km. But two out of three paragraphs are totally citations of opinions from previous papers. (a) Consider reducing the citation and increasing the ratio of analysis based on figures of the current paper. (b) Please add information to describe the wave here. Before using gravity wave acceleration to describe the intensity variation of the wave, the authors should first demonstrate the existence of gravity wave by showing amplitude or phase of the wave. Response: Temperature fluctuation is the main reason for the formation of 9-12 km cirrus cloud. Convective activities are important sources of gravity waves, which is responsible for inducing temperature fluctuations. Minor comments: 1. P1L21 needs to be rephrased, consider changing to "the cirrus clouds with different cloud top heights exhibit obvious difference in their horizontal distribution over the TP..." Response: Thank you for the suggestion. This part has been rephrased. 2. P2L1 "the maximum occurrence for cirrus top below 9 km . . ." is confusing in ex- pression. "Cirrus" or "cirrus with cloud tops in range . . ."? please clarify this concept and rephrase the sentence. Response: Thank you for the suggestion. This part has been rephrased as "The maximum occurrence for cirrus with cloud top height less than 9 km . . .". 3. P5L17 ". . .from 25° - 45°N and 65° - 105°E. . .": change to "the TP is defined as the area that covers25° - 45°N and 65° - 105°E" or similar expression. Response: Thank you for the suggestion. This part has been rephrased. 4. P7L14-15 and P821: leave a space between the number and the unit "km", as i.e., "9 km". Response: All the height description without a space between the number and the unit has been corrected. 5. Is it better to replace the colormap used for Figs 1-3a with a sequential or miscellaneous colormap? The current diverging one is not appropriate for describing monotonic trends. Besides, the font size of the "cirrus occurrence number" below

the colorbar should be enlarged. Response: Thank you for the suggestion. All the figures have been replotted. We appreciate Reviewer 2 very much for his constructive comments.

Please also note the supplement to this comment:
https://www.atmos-chem-phys-discuss.net/acp-2019-1000/acp-2019-1000-AC3-supplement.pdf
* * *
[Figure]

**Fig. 1.** the zonal distribution of vertical winds averaged from 80E to 90E for each latitude. The contour is specific humidity.

---

## Author Response (AR2)

Qianshan He
Shanghai Meteorological Service
oxeye75@163.com
July 30, 2020

Editor-in-Chief
Atmospheric Chemistry and Physics
Dear Editor:

Please find the revised version of "Possible mechanisms of summer cirrus clouds over the Tibetan Plateau". We value the comments received greatly and have accepted and incorporated essentially all of the reviewers' suggestions into the manuscript. The point-by-point replies to the reviewers' comments are enclosed.

We appreciate you very much for your editorial effort to this manuscript.

Sincerely yours,
Qianshan He

1.  This concise paper describes the feature of the summer cirrus clouds over the Tibetan plateau, particularly the cirrus top heights. Three potential formation mechanisms which determine the cirrus top height over the Tibetan Plateau have been proposed and evidence has been discussed. While I wonder if there are other potential mechanisms which also play a role (I think there might be), this study provides valuable information and results regarding the characteristics and formation of cirrus clouds over Tibetan Plateau. Thus, I would recommend its consideration for publication after some minor changes. 2. Zhang et al. "Possible mechanisms of summer cirrus clouds over the Tibetan Plateau"

Most references used in this study are those before 2013, some recent studies are worthy to review and cite.

**Response:** Thank you for the suggestion. Yes, the formation of cirrus requires joint efforts of sufficiently cold and moist atmospheric conditions, favorable convective activities as well as possible condensation nucleus. Other potential mechanisms, such as the Rossby wave (Dai, Wu, Song, & Liu, 2018) could also play a role. Our study tends to address the most dominating mechanisms to generate cirrus over the Plateau and provide an insight into their physical process. Therefore, other relatively trivial and less significant mechanisms are ignored in this paper. In the future, we will explore other mechanisms more thoroughly by case study if possible.

The above statements have been added at line 227-232 as well as the references.

References after 2013 have been cited.

Line 17-25, Regarding the abstract, a little more information about the relative importance of the three mechanisms at different locations/conditions would be helpful.

**Response:** The abstract has been modified at Line 24-26.

Line 29-32, You may rephrase this sentence a little bit.

**Response:** The sentence has been rephrased at Line 31-32.

Line 41-43, Two other recent references could be also cited here, which are Yang et al. (202, doi: 10.1016/j.atmosres.2020.104927) and Zhao et al. (2019, DOI:10.1002/joc.5975). They found that associated with the topographic effects and the transport of moisture air, the cirrus clouds are found more from March to May.

**Response:** The references are added.

Line 56-60, when we talk about the occurrence, characteristics and potential causes form ground-based observations, we usually took use of radar and lidar observations, such as Zhao et al. (2016, http://dx.doi.org/10.1175/JAMC-D-16-0038.1) for ice clouds and (2017, doi: 10.1016/j.atmosres.2017.02.002) for liquid clouds over the Tibetan Plateau. However, we should note that when using lidar observations, it is rare to examine them based on top height due to the attenuation of lidar signals, unless the clouds are thin without any other clouds below. Actually, with strong attenuation ability, radar (such as Zhao et al., 2016) can provide the characteristics (temporal and spatial distribution) of ice clouds over the Tibetan Plateau.

**Response:** Thanks for the comments. The statements have been added at line 56-62 as well as the references.

Line 75-76, we know that the third Tibetan Plateau Atmospheric Experiment was carried out in 2014 and 2015, it might be valuable and interesting to compare the findings from Calipso with that from ground-based remote sensing observations.

**Response:** Thanks for the comments. Our study is consistent with the third Tibetan Plateau Atmospheric Experiment both in terms of cirrus height (9-16 km) as well as the mechanism (deep convection). The related statements are added in Line 118-120 and Line 167.

Another question, Calipso observation only occurs at 1:30 PM in a 16-day cycle, how could this represent the characteristics of cirrus for a day. As Wang and Zhao (2017,

doi:10.1002/2016JD025954) indicated, even for MODIS which pass two times a day, the representation time error due to single time observation could be large when considering daily average or even monthly average.

**Response:** Our study is based on the distribution of the total cirrus occurrence numbers detected by CALIPSO during the June-August period from 2012-2016. It is the climatology results instead of the daily or monthly average. The larger the cirrus occurrence number, the more cirrus are guaranteed at least during the 1:30 PM local time. Yes, our study is somehow deficient due to the sampling time, but the conclusions we draw from the 5-year data can still represent the characteristics of cirrus over the entire Tibetan Plateau to a certain extent.

Line 93, Using superscript for Wm-2

**Response:** Corrected.

Line 117, large number of

**Response:** Corrected.

Line 125-126, actually, the heat pump effect could be further enhanced by the absorption of solar radiation by transported absorbing aerosols such as black carbon and dust, as indicated by zhao et al. (2020, https://doi.org/10.1093/nsr/nwz184).

**Response:** The comments have been added at Line 124-126.

Line 128-129, I am not sure if it is possible to get the radiation balance from CERES. If it is, you may check the consistency between CERES observations and ERA5.

**Response:** CERES provides radiation observation. The global ERA-Interim monthly surface incident shortwave radiation product had an overall correlation coefficient of 0.95, a bias of 11.25 $Wm^{-2}$ and an RMSE of 27.70 $W\ m^{-2}$ by comparing with the CERES data (Zhang et al., 2016). ERA5 has better performance with reduced errors (ERA5- CERES) than ERA-Interim did (Hogan et al., 2017). Therefore, ERA5 radiation product is reliable. Our study aims to compare the radiation cooling with latent heat, sensible heat and evaporation. It would be

more consistent if we use the same reanalysis product. CERES alone can only provide the radiation date. The possible inconsistency between CERES and ERA5 can bring bias to our analysis when we try to compare radiation with other physical processes.

Line 134, "Fig. 3b and Fig. 3c show the …"

**Response:** Corrected.

Line 140-141, is this a typo "with terrain height larger than 4500 km"?

**Response:** Corrected as "with terrain height larger than 4500 m". The sentence is rephrased at Line 145-146.

Line 144, you may rephrase this sentence to make it more clear "Still, the weak vertical motion in the upper layers prohibits the vertical growth of cirrus".

**Response:** The sentence is rephrased as "the weak vertical motion above 300 hPa further prohibits the vertical growth of cirrus to a larger height" at Line 146-147

Dai, G., Wu, S., Song, X., & Liu, L. (2018). *Optical and geometrical properties of cirrus clouds over the Tibetan Plateau measured by lidar and radiosonde sounding at the summertime in 2014.* Paper presented at the Optics and Photonics for Energy and the Environment.

Hogan, R. J., Ahlgrimm, M., Balsamo, G., Beljaars, A., Berrisford, P., Bozzo, A., . . . Lang, S. (2017). *Radiation in numerical weather prediction*: European Centre for Medium-Range Weather Forecasts.

Zhang, X., Liang, S., Wang, G., Yao, Y., Jiang, B., & Cheng, J. (2016). Evaluation of the reanalysis surface incident shortwave radiation products from NCEP, ECMWF, GSFC, and JMA using satellite and surface observations. *Remote Sensing, 8*(3), 225.